# LATENT PROGRAMMER: DISCRETE LATENT CODES FOR PROGRAM SYNTHESIS

## ABSTRACT

In many sequence learning tasks, such as program synthesis and document summarization, a key problem is searching over a large space of possible output sequences. We propose to learn representations of the outputs that are specifically meant for search: rich enough to specify the desired output but compact enough to make search more efficient. Discrete latent codes are appealing for this purpose, as they naturally allow sophisticated combinatorial search strategies. The latent codes are learned using a self-supervised learning principle, in which first a discrete autoencoder is trained on the output sequences, and then the resulting latent codes are used as intermediate targets for the end-to-end sequence prediction task. Based on these insights, we introduce the *Latent Programmer*, a program synthesis method that first predicts a discrete latent code from input/output examples, and then generates the program in the target language. We evaluate the Latent Programmer on two domains: synthesis of string transformation programs, and generation of programs from natural language descriptions. We demonstrate that the discrete latent representation significantly improves synthesis accuracy.

## 1 INTRODUCTION

Our focus in this paper is program synthesis, one of the longstanding grand challenges of artificial intelligence research (Manna & Waldinger, 1971; Summers, 1977). The objective of program synthesis is to automatically write a program given a specification of its intended behavior, such as a natural language description or a small set of input-output examples. Search is an especially difficult challenge within program synthesis (Alur et al., 2013; Gulwani et al., 2017), and many different methods have been explored, including top-down search (Lee et al., 2018), bottom up search (Udupa et al., 2013), beam search (Devlin et al., 2017), and many others (see Section 2).

We take a different philosophy: *Can we learn a representation of programs specifically to help search?* A natural way of representing a program is as a sequence of source code tokens, but the synthesis task requires searching over this representation, which can be difficult for longer, more complex programs. A programmer often starts by specifying high-level components of a program as a plan, then fills in the details of each component i.e. in string editing, a plan could be to extract the first name, then the last initial. We propose to use a sequence of latent variable tokens, called *discrete latent codes*, to represent such plans. Instead of having a fixed dictionary of codes, we let a model discover and learn what latent codes are useful and how to infer them from specification.

Our hypothesis is that a discrete latent code – a sequence of discrete latent variables – can be a useful representation for search (van den Oord et al., 2017; Roy et al., 2018; Kaiser et al., 2018). This is because we can employ standard methods from discrete search, such as beam search, over a compact space of high-level plans and then over programs conditioned on the plan, in a two-level procedure. We posit that the high-level search can help to organize the search over programs. In the string editing example earlier, a model could be confident that it needs to extract the last initial, but is less sure about whether it needs to extract a first name. By changing one token in the latent code, two-level search can explore alternative programs that do different things in the beginning. Whereas in traditional single-level search, the model would need to change multi-token prefixes of the alternatives, which is difficult to achieve in limited budget search.

We propose the *Latent Programmer*, a program synthesis method that uses learned discrete representations to guide search via a two-level synthesis. The Latent Programmer is trained by a self-

| Inputs | Outputs | Program |
|--------|---------|---------|
| "Mason Smith" | "Smith M" | |
| "Henry Myers" | "Myers H" | `GetToken_PROP_CASE_2 | Const(" ") |` |
| "Barry Underwood" | "Underwood B" | `GetToken_ALL_CAPS_1` |
| "Sandy Jones" | "Jones S" | |

Figure 1: A string transformation task with 4 input-output examples a possible program in the string transformation DSL that is consistent with the examples.

supervised learning principle. First a discrete autoencoder is trained on a set of programs to learn discrete latent codes, and then an encoder is trained to map the specification of the synthesis task to these latent codes. Finally, at inference time, Latent Programmer uses a two-level search. Given the specification, the model first produces a $L$-best list of latent codes from the latent predictor, and uses them to synthesize potential programs. On two different program synthesis domains, we find empirically that the Latent Programmer improves synthesis accuracy by over $10\%$ compared to standard sequence-to-sequence baselines as RobustFill (Devlin et al., 2017). We also find that our method improves diversity of predictions, as well as accuracy on long programs.

## 2 BACKGROUND

**Problem Setup**  The goal in program synthesis is to find a program in a given language that is consistent with a specification. Formally, we are given a domain specific language (DSL) which defines a space $\mathcal{Y}$ of programs. The task is described by a specification $X \in \mathcal{X}$ and is solved by some, possibly multiple, unknown program(s) $Y \in \mathcal{Y}$. For example, each specification can be a set of input/output (I/O) examples denoted $X = \{(I_1, O_1), \ldots (I_N, O_N)\}$. Then, we say that we have solved specification $X$ if we found a program $Y$ which correctly solves all the examples: $Y(I_i) = O_i, \forall i = 1, \ldots, N$. As another example, each specification can be a natural language description of a task, and the corresponding program implements said task. An example string transformation synthesis task with four I/O examples together with a potential correct program in the string transformation DSL is shown in Figure 1.

**Vector Quantization**  Traditionally, neural program synthesis techniques process the input specification as a set of sequences and predicts the output program token-by-token (Devlin et al., 2017). In this work, we present a new approach for synthesis that performs structured planning in latent space using a discrete code. We conjecture that programs have an underlying discrete structure; specifically, programs are compositional and modular with components that get reused across different problems. Our approach leverages this structure to guide the search over large program spaces. Following works in computer vision (van den Oord et al., 2017; Roy et al., 2018), we discover such discrete structure by using a Vector Quantized Variational Autoencoder (VQ-VAE). VQ-VAEs work by feeding the intermediate representation of an autoencoder through a discretization bottleneck (van den Oord et al., 2017). For completeness, we provide background on VQ-VAEs below.

In a VQ-VAE, latent codes are drawn from a discrete set of learned vectors $c \in \mathbb{R}^{K \times D}$, or codebook. Each element in the codebook can be viewed as either a token with id $k \in [K]$ or as an embedding $c_k \in \mathbb{R}^D$. To generate the discrete codes, the continuous autoencoder output $e$ is quantized via nearest-neighbor lookup into the codebook. Formally, the token id $\mathrm{qk}(e)$ and quantized embedding $\mathrm{qc}(e)$ are defined as

$$\mathrm{qc}(e) = c_{\mathrm{qk}(e)} \text{ where } \mathrm{qk}(e) = \arg \min_{k \in [K]} ||e - c_k||_2. \tag{1}$$

For input $x$, the training loss for a VQ-VAE consists of: a reconstruction loss for the encoder-decoder weights, a codebook loss that encourages codebook embeddings to be close to the continuous vectors which are quantized to them, and a commitment loss that encourages the encoded input $\mathrm{ec}(x)$ to "commit" to codes i.e. not switch which discrete code it is quantized to. The loss is given by,

$$\mathcal{L}(c, \theta, \phi) = \log p_\theta \left( x \mid \mathrm{qc}(\mathrm{ec}_\phi(x)) \right) + ||\mathrm{sg}(\mathrm{ec}_\phi(x)) - c||_2^2 + \beta ||\mathrm{sg}(c) - \mathrm{ec}_\phi(x)||_2^2, \tag{2}$$

where $\theta, \phi$ are the parameters of the decoder and encoder, respectively, $\mathrm{sg}(\cdot)$ is the stop gradient operator that fixes the operand from being updated by gradients, and $\beta$ controls the strength of the commitment loss. To stabilize training, van den Oord et al. (2017) also proposed removing the codebook loss and set the codebook to an exponential moving average (EMA) of encoded inputs.

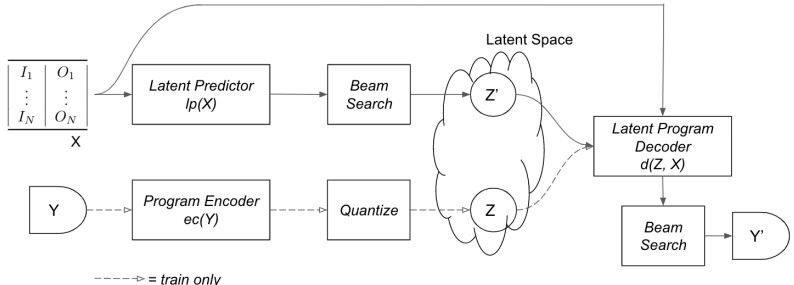

Figure 2: High-level architecture for the Latent Programmer system. The latent predictor generates probabilities over latent sequences, which can be decoded into a predicted latent sequence $Z'$. $Z'$ is fitted to a ground-truth latent sequence $Z$ generated by a program encoder, and used during decoding to by the latent program decoder to generate programs.

## 3 Synthesis with Discrete Latent Variables

We propose a two-level hierarchical approach to program synthesis that first performs high-level planning over an intermediate sequence, which is then used for fine-grained generation of the program. In our approach, a top-level module first infers a latent code, which gets used by a low-level module to generate the final program.

### 3.1 Hierarchy of Two Transformers

Our proposed **Latent Programmer** (LP) architecture consists of two Transformers in a two-level structure. The architecture comprises of two modules: a **latent predictor** which produces a latent code, which can be interpreted as a course sketch of the program, and a **latent program decoder**, which generates a program conditioned on the code. The latent code consists of discrete latent variables as tokens, which we arbitrarily denote `TOK_1,...,  TOK_K`, whose meanings are assigned during training. Both components use a Transformer architecture due to their impressive performance on natural language tasks (Vaswani et al., 2017).

To help the model assign useful meanings to the latents, we also leverage a **program encoder**, which is only used during training. The program encoder $\mathrm{ec}(Y)$ encodes the true program $Y = [y_1, y_2, \ldots, y_T]$ into a shorter sequence of discrete latent variables $Z = [z_1, z_2, \ldots, z_S]$, represented as codebook entries; that is, each $z_i \in \mathbb{R}^D$ is one of $K$ entries in a codebook $c$. The latent sequence serves as the ground-truth high-level plan for the task. The function $\mathrm{ec}(Y)$ is a Transformer encoder, followed by a stack of convolutions of stride 2, each halving the size of the sequence. We apply the convolution $\ell$ times, which reduces a $T$-length program to a latent sequence of length $\lceil T/2^\ell \rceil$. This provides temporal abstraction, since the high-level planning actions are made only every $2^\ell$ steps. In summary, the program encoder is given by

$$\mathrm{ec}(Y) \leftarrow h_\ell; \quad h_m \leftarrow \mathrm{Conv}(h_{m-1}) \text{ for } m \in 1 \ldots \ell; \quad h_0 \leftarrow \mathrm{TransformerEncoder}(Y). \quad (3)$$

Here $\mathrm{TransformerEncoder}(\cdot)$ applies a stack of self-attention and feed-forward units on input embeddings via a residual path, described in detail by Vaswani et al. (2017). This will be used, along with the latent program decoder, as an autoencoder during training (see Section 3.2).

The **latent predictor** $\mathrm{lp}(X)$ autoregressively predicts a coarse latent code $\mathrm{lp}(X) \in \mathbb{R}^{S \times K}$, conditioned on the program specification $X$. The latent predictor outputs a sequence of probabilities, which can be decoded using search algorithms such as beam search to generate a predicted latent code $Z'$. This is different than the program encoder, which outputs a single sequence $Z$, because we use the latent predictor to organize search over latent codes; at test time, we will obtain a $L$-best list of latent token sequences from $\mathrm{lp}(X)$. The latent predictor is given by a stack of Transformer blocks with the specification $X$ as inputs.

Similarly, the **latent program decoder** $d(Z, X)$ defines an autoregressive distribution over program tokens given the specification $X$ and the coarse plan $Z \in \mathbb{R}^{S \times K}$, represented as codebook entries. The decoder is a Transformer that jointly attends to the latent sequence and program specification. This is performed via two separate attention modules, whose outputs are concatenated into the hidden unit. Formally, given a partially generated program $Y' = [y'_1, y'_2, \ldots, y'_{t-1}]$, and the encoded

specification $E = \text{TransformerEncoder}(X)$, the latent program decoder performs

$$h_t = \text{Concat}\left(\text{TransformerDecoder}(Y', E)_{t-1}, \text{TransformerDecoder}(Y', Z)_{t-1}\right), \quad (4)$$

where $\text{TransformerDecoder}(x, y)$ denotes a Transformer decoder applied to outputs $y$ while attending to inputs encoding $x$, and the subscript indexes an entry in the resulting output sequence. Finally, the distribution over output token $k$ is given by $d_t(Z, X) = \text{Softmax}\left(W(h_t)\right)$, where $W$ is a learned parameter matrix. Finally, the latent program decoder defines a distribution over programs autoregressively as $p(Y|Z, X) = \prod_t p(y_t | y_{<t}, Z, X)$, where $p(y_t | y_{<t}, Z, X) = d_t(Z, X)$. When $X$ is multiple I/O examples, each example is encoded as $E_i = \text{TransformerDecoder}(I_i, O_i)$. Then, a separate hidden state per I/O is computed following equation 4, followed by a late max-pool to get the final hidden state. Note that the program encoder and latent program decoder make up a VQ-VAE model of programs, with additional conditioning on the specification.

The complete LP architecture is summarized in Figure 2, and an end-to-end example run of our architecture is shown in Figure 4.

## 3.2 TRAINING

Our LP performs program synthesis using a two-level search, first over latent sequences then over programs. Given program specification, we want to train our latent predictor to produce an informative latent sequence from which our latent program decoder can accurately predict the true program. Our training loss for the LP model consists of three supervised objectives.

The **autoencoder loss** ensures that the latent codes contain information about the program. It is a summation of the reconstruction loss between the autoencoder output $d(\text{qc}(Y), X)$ and true program $Y$, as well as a commitment loss to train the encoder output $\text{ec}(Y)$ to be close to codebook $c$. Like in Roy et al. (2018), codebook is not trained but set to the EMA of encoder outputs. This loss is similar to the loss function of a VQ-VAE as in equation 2, but also depends on specification $X$. This objective trains the latent tokens in the codebook so that they correspond to informative high-level actions, as well as make sure our latent program decoder can accurately recover true program given the specification and a plan comprising of such actions.

The **latent prediction loss** ensures that latent codes can be predicted from specifications. It is a reconstruction loss between the distribution over latents predicted from the specification $\text{lp}(X)$ and the autoencoded latents $\text{qk}(\text{ec}(Y))$ from the ground-truth program. This is a self-supervised approach that treats the autoencoded latent sequence as the ground-truth high-level plan, and trains the latent predictor to generate the plan using just the program specification $X$. Note that the program encoder is only used in training, as at test time $\text{ec}(Y)$ is unknown, so the LP model uses $\text{lp}(X)$ instead.

Finally, the **end-to-end loss** ensures that programs can be predicted from specifications. This is especially important because in the reconstruction loss, the latent program decoder receives as input latent codes from the autoencoded latent sequences $\text{ec}(Y)$, whereas at test time, the decoder receives a latent code from the latent predictor $\text{lp}(X)$. This can result in mistakes in the generated program since the decoder has never been exposed to noisy results from the latent predictor. The end-to-end loss alleviates this issue. The end-to-end loss is probability of the correct program $Y$ when predicted from a soft-quantized latent code, given by $\text{lp}(X)^T c$. This has the added benefit of allowing gradient to flow through the latent predictor, training it in an end-to-end way.

In summary, the full loss for a training instance is

$$\mathcal{L}(c, \theta, \phi, \psi) = \underbrace{\log p_\theta\left(Y \mid \text{qc}(\text{ec}_\phi(Y)), X\right) + \beta||\text{sg}(c) - \text{ec}_\phi(Y)||_2^2}_{\text{autoencoder}}$$

$$+ \underbrace{\log p\left(\text{qk}(\text{ec}_\phi(Y)) \mid \text{lp}_\psi(X)\right)}_{\text{latent prediction}} + \underbrace{\log p_\theta\left(Y \mid \text{lp}_\psi(X)^T c, X\right)}_{\text{end-to-end}} \quad (5)$$

where we explicitly list out $\theta, \phi$, and $\psi$ representing the parameters of the latent program decoder, program encoder, and latent decoder respectively.

Furthermore, for the first 10K steps of training, we give embeddings of the ground-truth program $Y$, averaged over every $2^\ell$ tokens, as the latent sequence instead of $\text{ec}(Y)$. This pre-training ensures that initially, the latent code carries some information about the program so that the attention to the code

has reasonable gradients that can then to propagated to the program encoder afterward pre-training. Doing this was empirically shown to prevent the bypassing phenomenon where the latent code is ignored during decoding (Bahuleyan et al., 2017).

### 3.3 INFERENCE

During inference, we use a multi-level variant of beam search to decode the output probabilities of our LP model. Standard beam search with beam $B$ will generate the top-$B$ most likely programs according to the model, and find the first one (if any) that is consistent with the specification (Parisotto et al., 2017; Devlin et al., 2017). In our case, we first perform beam search for $L$ latent beams, then for $\lfloor B/L \rfloor$ programs per latent sequence. Note that during inference, the latent predictor will continue to generate latent tokens until an end-of-sequence token is produced. This means that the generated latent sequence does not necessarily satisfy having length $\lceil T/2^\ell \rceil$ as during training; however, we found the latent sequence lengths during training and evaluation to be close in practice. Setting $L = B$ allows for the maximum exploration of the latent space, while setting $L = 1$ reduces our method to standard beam search, or exploitation of the most likely latent decoding. We choose $L = \sqrt{B}$ in our experiments, but explore the effect of various choices of $L$ in Section 5.2.

## 4 RELATED WORK

**Program Synthesis**    Our work deals with *program synthesis*, which involves combinatorial search for programs that match a specification. Many different search methods have been explored within program synthesis, including search within a version-space algebra (Gulwani, 2011), bottom-up enumerative search (Udupa et al., 2013), stochastic search (Schkufza et al., 2013), genetic programming (Koza, 1994), or reducing the synthesis problem to logical satisfiability (Solar-Lezama et al., 2006). *Neural program synthesis* involves learning neural networks to predict function distributions to guide a synthesizer (Balog et al., 2017), or the program autoregressively in an end-to-end fashion (Parisotto et al., 2017; Devlin et al., 2017). SketchAdapt (Nye et al., 2019) combined these approaches by first generating a program sketch with holes, and then filling holes using a conventional synthesizer. Related to our work, DreamCoder (Ellis et al., 2020) iteratively builds a sketches using progressively more complicated primitives though a wake-sleep algorithm. Our work is closely related in spirit but fundamentally differs in two ways: (1) our sketches are comprised of a general latent vocabulary that is learned in a simple, self-supervised fashion, and (2) our method avoids enumerative search, which is prohibitively expensive for large program spaces. There is also a line of work that deals with learning to process partial programs in addition to the specification. In *execution-guided program synthesis*, the model guides iterative extensions of the partial programs until a matching one is found (Zohar & Wolf, 2018; Chen et al., 2019; Ellis et al., 2019). Balog et al. (2020) of late proposed a differentiable fixer that is trained to iteratively edit incorrect programs. We treat these works as complementary, and can be combined with ours to refine predictions.

**Discrete Latent Bottlenecks**    Variational autoencoders (VAE) were first introduced using continuous latent representations (Kingma & Welling, 2014; Rezende et al., 2014). Several promising approaches were proposed to use discrete bottlenecks instead, such as continuous relaxations of categorical distributions i.e. the Gumbel-Softmax reparametrization trick (Jang et al., 2017; Maddison et al., 2017). Recently, VQ-VAEs using nearest-neighbor search on a learned codebook (see Section 2 for more details) achieved impressive results almost matching continuous VAEs  (van den Oord et al., 2017; Roy et al., 2018). Discrete bottlenecks have also been used for sentence compression (Miao & Blunsom, 2016) and text generation (Puduppully et al., 2019), but these works does not learn the semantics of the latent codes, like ours does. Within the domain of synthesis of chemical molecules, Gómez-Bombarelli et al. (2018) have applied Bayesian optimization within a continuous latent space to guide this structured prediction problem. Learning to search has also been considered in the structured prediction literature (Daumé et al., 2009; Chang et al., 2015; Ross et al., 2011), but to our knowledge, these works do not consider the problem of learning a discrete representation for search. Notably, VQ-VAE methods have been successfully used to encode natural language into discrete codes for faster decoding in machine translation (Kaiser et al., 2018). Our work similarly uses a VQ-VAE to learn a discrete code, but we use the learned code in a two-level search that improves accuracy. To do so, we propose a model that is autoregressive on both the latent and program space, and perform two-level beam search on latent codes and programs. The key novelty behind our work is that first searching over a learned discrete latent space can assist search over the complex program space; using a VQ-VAE as Kaiser et al. (2018) did enables us to do so.

## 5 EXPERIMENTS

We now present the results of evaluating our Latent Programmer model in two test domains: synthesis of string transformation programs from examples and code generation from natural language descriptions. We compare our LP model against several strong baselines.

**RobustFill [LSTM]** is a seq-to-seq LSTM with attention on the input specification, and trained to autoregressively predict the true program. The architecture is comparable to the RobustFill model designed originally for the string transformation tasks in our first domain (Devlin et al., 2017), but easily generalizes to all program synthesis domains. We detail the architecture in Appendix A.

**RobustFill [Transformer]** alternatively uses a Transformer architecture, equivalent in architecture to the latent planner in our LP model, also trained to autoregressively predict the program. Transformers were found to perform much better than LSTMs in language tasks because they process the entire input as a whole, and have no risk of forgetting past dependencies (Vaswani et al., 2017). This baseline can be also be considered of an ablation of our LP model without any latent codes.

The central novelty of our work is in realizing that by learning a discrete representation, we can perform structured search on two levels. We introduce two ablative baselines, which replace the VQ-VAE with either a generic autoencoder or a VAE. In both cases the latent space is continuous, and well-known combinatorial search algorithms such as beam search cannot search over the space.

**Latent RobustFill [AE]** replaces the VQ-VAE component of our LP model with a generic autoencoder. This makes the latent code a sequence of continuous embeddings. The latent prediction loss in equation 5 is simply replaced by a squared error between the output of the autoencoder and the latent predictor. Performing beam search over the continuous latent space is intractable, so during inference we generate only one latent sequence per task; this is equivalent to two-level beam search described earlier with $L = 1$. In addition, because we cannot define an end-of-sequence token in the latent space, this baseline must be given knowledge of the true program length even during inference, and always generates a latent sequence of length $\lceil T/2^{\ell} \rceil$.

**Latent RobustFill [VAE]** substitutes the VQ-VAE component with a VAE (Kingma & Welling, 2014). This again produces a continuous latent space, but regularized to be distributed approximately as a standard Gaussian. Performing beam search is still intractable, but we can sample $L$ latent sequences from the Gaussians determined by the VAE, and perform beam search on the programs afterwards. Again, we assume that the true program length is known during inference.

### 5.1 STRING TRANSFORMATION

The first test domain is a string transformation DSL frequently studied in the program synthesis literature (Parisotto et al., 2017; Devlin et al., 2017; Balog et al., 2020). Tasks in this domain involve finding a program which maps a set of input strings to a corresponding set of outputs. Programs in the DSL are a concatenation of expressions that perform regex-based string transformations (see Appendix A for the full DSL).

We perform experiments on a synthetic dataset generated by sampling programs from the DSL, then the corresponding I/O examples using an heuristic similar to the one used in NSPS (Parisotto et al., 2017) and RobustFill (Devlin et al., 2017) to ensure nonempty output for each input. We consider programs comprising of a concatenation of up to 10 expressions and limit the lengths of strings

| Method | Accuracy | | |
|---|---|---|---|
| | B = 1 | 10 | 100 |
| RobustFill [LSTM] | 45% | 49% | 61% |
| RobustFill [Transformer] | 47% | 51% | 61% |
| Latent RobustFill [AE] | 47% | 50% | 60% |
| Latent RobustFill [VAE] | 46% | 51% | 62% |
| Latent Programmer | **51%** | **57%** | **68%** |

Table 1: Accuracy on string transformation domain.

in the I/O to be at most 100 characters. All models have an embedding size of 128 and hidden size of 512, and the attention layers consist of 3 stacked layers with 4 heads each. For the LP model, we used a latent compression factor $\ell = 2$ and vocabulary size $K = 40$. The models are trained on roughly 25M tasks, and evaluated on 1K held-out ones.

In Table 1, we report the accuracy–the number of time a program was found conforming to the I/O examples–of our method against the baselines. Across all beam sizes, our LP model performed 5-7

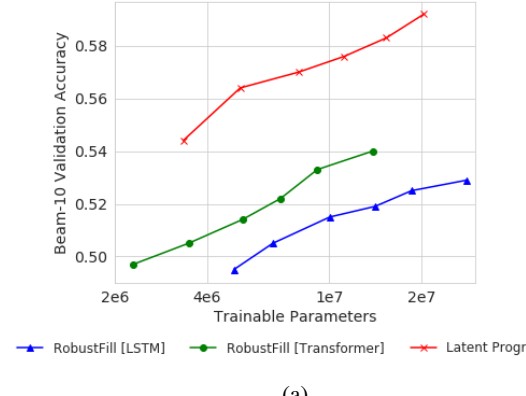

| Length | RobustFill Acc. | LP Acc. |
|--------|-----------------|---------|
| 1 | **94.5**% | 94.0% |
| 2 | 83.9% | **84.6**% |
| 3 | **72.8**% | 72.2% |
| 4 | 63.1% | **66.1**% |
| 5 | 47.1% | **49.8**% |
| 6 | 40.6% | **43.0**% |
| 7 | 30.2% | **34.6**% |
| 8 | 22.7% | **28.4**% |
| 9 | 18.6% | **27.0**% |
| 10 | 14.4% | **25.6**% |

(a)   (b)

Figure 3: (a): Influence of hidden size on beam-10 accuracy. (b): Beam-10 accuracy of baseline transformer and LP by ground truth program length.

percentage points better (over 10% of baseline accuracy) than the next best model. From our ablative study, we see that having two-level using discrete latent codes was important, as the baselines over continuous latent spaces performed comparably to the traditional RobustFill model.

Recently, SketchAdapt also proposed two-level search (Nye et al., 2019), but in the top-level, it performs beam search over program space augmented with a `HOLE` token. In contrast, our method searches over a learned, general latent space. During low-level search, SketchAdapt enumerates partial programs to co-opt the `HOLE` tokens using a learned syn-

| Method | Accuracy | |
|--------|----------|---|
| | B=50 | 100 |
| SketchAdapt (Nye et al., 2019) | 63% | 64% |
| Latent Programmer | **64**% | **67**% |

Table 2: Accuracy on string transformation domain of Nye et al. (2019). SketchAdapt results are from Nye et al. (2019) with $3,000$ synthesized programs (similar wall clock time).

thesizer similar to DeepCoder (Balog et al., 2017), whereas we again perform beam search. To compare the two, we evaluate our LP model on samples generated according to Nye et al. (2019), which slightly modifies the DSL to increase the performance of synthesizers, and report results in Table 2. Since enumeration can be done more quickly than beam search, we let SketchAdapt synthesize $3,000$ programs using $B$ top-level beams, whereas our LP model can only generate $B$ programs. Our LP model is able to outperform SketchAdapt even in the modified DSL.

## 5.2 ANALYSIS

We conduct extensive analysis to better understand our LP model in terms of learning, the ability to generate long programs, and diversity in the beams. All results are reported with beam size $B = 10$.

**Model Size**   Our LP model uses an additional latent code for decoding, which introduces additional parameters into the model than the baseline RobustFill model. To make a fair comparison, we vary the embedding and hidden dimension of all of our evaluated methods, and compare the effect of the number of trainable parameters on the accuracy. Figure 3(a) shows that all methods respond well to an increase in model size. Nevertheless, we see that even when normalized for size, our LP model outperforms baselines by a significant margin.

**Program Length**   Prior work has shown that program length is a reasonable proxy measure of problem difficulty. We hypothesize that using latent codes is most beneficial when generating long programs. Figure 3(b) shows how ground-truth program length affects the accuracy of our LP model compared to RobustFill, which lacks latent codes. As expected, accuracy decreases with problem complexity. Perhaps surprisingly, though, we see a large improvement in our LP model's ability to handle more complex problems. In Figure 4, we also show an illustrative example in the domain where our LP model found a valid program whereas the RobustFill model did not. In this example, the ground-truth program was long but had a repetitive underlying structure. Our LP model correctly detected this structure, as evidenced by the predicted latent sequence. We show additional examples

| Inputs | Outputs | Program |
|--------|---------|---------|
| "Jacob,Ethan,James 11" | "11:J.E.J." | `GetToken_NUMBER_1   \| Const(:)   \|` |
| "Elijah,Daniel,Aiden 3162" | "3162:E.D.A" | `GetToken_ALL_CAPS_1 \| Const(.)   \|` |
| "Rick,Oliver,Mia 26" | "26:R.O.M." | `GetToken_ALL_CAPS_2 \| Const(.)   \|` |
| "Mark,Ben,Sam 510" | "510:M.B.S." | `GetToken_ALL_CAPS_3 \| Const(.)` |

| | |
|---|---|
| RobustFill | `GetAll_NUMBER \| Const(:)\| GetToken_ALL_CAPS_2 \| Const(.)` |
| LP | `GetAll_NUMBER \| Const(:)  \| GetToken_ALL_CAPS_1 \| Const(.)   \|` `GetToken_ALL_CAPS_2 \| Const(.)  \| GetToken_ALL_CAPS_-1 \| Const(.)` |
| LP Latent | `TOK_14 \| TOK_36 \| TOK_36 \| TOK_36` |

Figure 4: Illustrative string transformation problem where the ground-truth program was long but had repetitive structure. The baseline Transformer was unable to generate the program but our LP model, which first predicts a coarse latent sequence, was able to.

| Latent Beam Size | Accuracy | Distinct n-Grams | | | |
|:---:|:---:|:---:|:---:|:---:|:---:|
| | | n = 1 | 2 | 3 | 4 |
| L = 1 | 52% | 0.13 | 0.23 | 0.26 | 0.28 |
| 2 | 55% | 0.13 | 0.24 | 0.26 | 0.28 |
| 3 | **57**% | 0.14 | 0.25 | 0.28 | 0.31 |
| 5 | 57% | 0.14 | 0.26 | 0.29 | 0.32 |
| 10 | 56% | **0.14** | **0.26** | **0.30** | **0.33** |

(a)

| $2^\ell$ | Accuracy |
|:---:|:---:|
| 2 | 52% |
| 4 | **55**% |
| 8 | 49% |

| $K$ | Accuracy |
|:---:|:---:|
| 10 | 48% |
| 40 | **55**% |
| 100 | 51% |

(b)

Figure 5: (a): Effect of latent beam size on beam-10 accuracy and number of distinct $n$-grams (normalized by total number of tokens). (b): Effect of latent length compression $\ell$ and vocabulary size $K$ on beam-10 accuracy.

in Figure 9 of Appendix B. It is important to note that our method allows tokens in the discrete latent code to have arbitrary meaning, yielding rich and expressive latent representations. However, the trade-off is that because the latent codes were not grounded, it is difficult to objectively interpret the latent codes. Grounding the latent space to induce interpretability is an avenue for future work.

**Latent Beam Size**   In multi-level beam search of beam size $B$, first $L$ latent beams are decoded, then $\lfloor B/L \rfloor$ programs per latent sequence. The latent beam size $L$ controls how much search is performed over latent space. We theorize that higher $L$ will produce more diverse beams; however, too high $L$ can be harmful in missing programs with high joint log-probability. We show the effect of latent beam size on both the beam-10 accuracy and a proxy measure for diversity. Following prior work, we measure diversity by counting the number of distinct $n$-grams in the beams, normalized by the total number of tokens to bias against long programs (Vijayakumar et al., 2018). We report the results varying $L$ for $B = 10$ in Figure 5(a). As expected, increasing the latent beam size $L$ improves diversity of output programs, but excessively large $L$ harms the final accuracy. An important observation is that the $L = 1$ case, where one latent code is used to decode all programs, performs similarly to baseline RobustFill. In this extreme, no search is performed over the latent space, and our proposed two-level search reduces to only searching over programs; this is further evidence that explicitly having two-level search is critical to the LP model's improved performance.

**Latent Length and Vocabulary Size**   Since the discretization bottleneck is a critical component in generating latent codes in our LP model, we also investigate its performance in conjunction with different settings of hyperparameters. Two important variables for the VQ are the latent length compression factor $c$, and size of latent vocabulary $K$. If $c$ is too small, the latent space becomes too large to search; on the other hard, too large $c$ can mean individual latent tokens cannot encoded the information needed to reconstruct the program. Similarly, we expect that too small of a vocabulary $K$ can limit the expressiveness of the latent space, but too large $K$ can make predicting the correct latent code too difficult. We confirm this in our evaluations in Figure 5(b) and Figure 5(c).

| Docstring | Program |
|---|---|
| get an environment variable | ```def getenv(key, default=None):```
```    return environ.get(key, default)``` |
| return a list of the words in the string s | ```def split(s, sep=None, maxsplit=-1):```
```    return s.split(sep, maxsplit)``` |

Figure 6: Example problems from the Python code generation dataset.

### 5.3 PYTHON CODE GENERATION

Our next test domain is a Python code generation (CG) task, which involves generating code for a function that implements a natural-language specification. The dataset used consists of 111K python examples, which consist of a docstring and corresponding code snippet, collected from Github (Wan et al., 2018). An example docstring and program from the dataset is shown in Figure 6.

We used a language-independent tokenizer jointly on data (Kudo & Richardson, 2018), and processed the dataset into a vocabulary of 35K sub-word tokens. Furthermore, following Wei et al. (2019), we set the maximum length of the programs to be 150 tokens resulting in 85K examples. Across all models, we set the embedding size to be 256 and hidden size to be 512, and the attention layers consist of 6 stacked layers with 16 heads each, similar to in neural machine translation (Vaswani et al., 2017). For the LP model, we used a latent compression factor $c = 2$ and vocabulary size $K = 400$ after a hyperparameter search. The models are evaluated on 1K held-out examples.

We initially found that it was difficult for the program encoder to detect latent sequence structure in the ground-truth programs as is due to the noise in variable names. To remedy this, we used an abstract syntax tree (AST) parser on the ground-truth programs to replace the $i$-th function argument and variable appearing the program with the token `ARG_i` and `VAR_i`, respectively. This was only used in training the program encoder and did not impact evaluation.

| Method | BLEU | | |
|---|---|---|---|
| | B = 1 | 10 | 100 |
| Base (Wei et al., 2019) | 10.4 | - | - |
| Dual (Wei et al., 2019) | 12.1 | - | - |
| RobustFill [LSTM] | 11.4 | 14.8 | 16.0 |
| RobustFill [Transformer] | 12.1 | 15.5 | 17.2 |
| Latent Programmer | **14.0** | **18.6** | **21.3** |

Table 3: BLEU score on code generation task.

We evaluate performance by computing the best BLEU score among the output beams (Papineni et al., 2002). We computed BLEU as the geometric mean of $n$-gram matching precision scores up to $n = 4$. Table 3 shows that our LP model outperforms the baselines. From the results, it can be seen that this is a difficult task, which may be due to the ambiguity in specifying code from a short docstring description. As evidence, we additionally include results from a recent work that proposed seq-to-seq CG models on the same data that performed similar to our baselines (Wei et al., 2019). These results show that improvements due to the LP model exist even in difficult CG domains. For example docstrings and code generated by the LP Model, refer to Figure 9 in Appendix B.

## 6 CONCLUSION

In this work we proposed the Latent Programmer (LP), a novel neural program synthesis technique that leverages a structured latent sequences to guide search. The LP model consists of a latent predictor, which maps the input specification to a sequence of discrete latent variables, and a latent program decoder that generates a program token-by-token while attending to the latent sequence. The latent predictor was trained via a self-supervised method in which a discrete autoencoder of programs was learned using a discrete bottleneck, specifically a VQ-VAE (van den Oord et al., 2017), and the latent predictor tries to predict the autoencoded sequence as if it were the ground-truth. During inference, the LP model first searches in latent space for discrete codes, then conditions on those codes to search over programs. Empirically, we showed that the Latent Programmer outperforms state-of-the-art baselines as Robustfill (Devlin et al., 2017), which ignore latent structure. Exciting future avenues of investigation include achieving better performance by grounding the latent vocabulary and generalizing our method to other tasks in natural language and structured prediction.

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

$$
\begin{array}{rcl}
\text{Program } Y & := & \texttt{Concat}(e_1, e_2, \ldots) \\
\text{Expression } e & := & f \mid n \mid n_1(n_2) \mid n(f) \mid \texttt{ConstStr}(c) \\
\text{Substring } f & := & \texttt{SubStr}(k_1, k_2) \mid \texttt{GetSpan}(r_1, i_1, b_1, r_2, i_2, b_2) \\
\text{Nesting } n & := & \texttt{GetToken}(t, i) \mid \texttt{ToCase}(s) \mid \texttt{Replace}(\delta_1, \delta_2) \mid \texttt{Trim}() \mid \texttt{GetUpto}(r) \mid \texttt{GetFrom}(r) \\
& & \mid \texttt{GetFirst}(t, i) \mid \texttt{GetAll}(t) \\
\text{Regex } r & := & t_1 \mid \ldots \mid t_n \mid \delta_1 \mid \ldots \mid \delta_m \\
\text{Type } t & := & \texttt{NUMBER} \mid \texttt{WORD} \mid \texttt{ALPHANUM} \mid \texttt{ALL\_CAPS} \mid \texttt{PROP\_CASE} \mid \texttt{LOWER} \mid \texttt{DIGIT} \mid \texttt{CHAR} \\
\text{Case } s & := & \texttt{PROPER} \mid \texttt{ALL\_CAPS} \mid \texttt{LOWER} \\
\text{Position } k & := & -100 \mid -99 \mid \ldots \mid 1 \mid 2 \mid \ldots \mid 100 \\
\text{Index } i & := & -5 \mid -4 \mid \ldots \mid 1 \mid 2 \mid \ldots \mid 5 \\
\text{Boundary } b & := & \texttt{START} \mid \texttt{END} \\
\text{Delimiter } \delta & := & \&, .?@()[]\%\{\}/ :; \$\#"' \\
\text{Character } c & := & A-Z \mid a-z \mid 0-9 \mid \&, .?@ \ldots
\end{array}
$$

Figure 7: The DSL for string transformation tasks (Devlin et al., 2017)

## A EXTENDED DESCRIPTION OF DSL AND ROBUSTFILL MODEL

The DSL for string transformations we use is the same as used in RobustFill (Devlin et al., 2017), and is shown in Figure 7. The top-level operator for programs in the DSL is a $\text{Concat}$ operator that concatenates a random number (up to 10) of expressions $e_i$. Each expression $e$ can either be a substring expression $f$, a nesting expression $n$, or a constant string $c$. A substring expression can either return the substring between left $k_1$ and right $k_2$ indices, or between the $i_1$-th occurence of regex $r_1$ and $i_2$-th occurence of regex $r_2$. The nesting expressions also return substrings of the input, such as extracting the $i$-th occurrence of a regex, but can also be composed with existing substring or nesting expressions for more complex string transformations.

**RobustFill Model** RobustFill (Devlin et al., 2017) is a seq-to-seq neural network that uses a encoder-decoder architecture where the encoder computes a representation of the input $e(X)$, and the decoder autoregressively generates the output given the source representation, i.e. conditional likelihood of $Y = [y_1, \ldots, y_T]$ decomposes as $p(Y|X) = \prod_{t=1}^{T} p(y_t|y_{<t}, X)$.

In RobustFill, the probability of decoding each token $y_t$ is given by $p(y_t|y_{<t}, X) = \text{Softmax}(W(h_t))$ with $W$ being the projection onto logits, or unnormalized log probabilities. The hidden representation $h_t$ is an LSTM hidden unit given by,

$$
E_t = \text{Attention}(h_{t-1}, e(X)),
$$
$$
h_t = \text{LSTM}(h_{t-1}, E_t).
$$

Here $e(X)$ is the sequence of hidden states after processing the specifications with an LSTM encoder, and $\text{Attention}(Q, V)$ denotes the scaled dot-product attention with query $Q$ and key-value sequence $V$ (Bahdanau et al., 2016). In the case of $X$ being multiple I/O examples, the RobustFill model of Devlin et al. (2017) uses double attention

$$
s_{t,i}^{I} = \text{Attention}(h_{t-1}, e(I_i))
$$
$$
s_{t,i}^{O} = \text{Attention}(\text{Concat}(h_{t-1}, s_{t,i}^{I}), e(O_i))
$$
$$
h_{t,i} = \text{LSTM}(h_{t-1}, \text{Concat}(s_{t,i}^{I}, s_{t,i}^{O})) \quad \forall 1 \le i \le N,
$$

and hidden states are pooled across examples before being fed into the final softmax layer, or $h_t = \text{maxpool}_{1 \le i \le N} \tanh(V(h_{t,i}))$, where $V$ is another projection.

## B   EXAMPLES OF GENERATED PROGRAMS AND LATENT CODES

| Inputs | Outputs | LP Outputs |
|---|---|---|
| "Mason Smith" | "Smith M" | "Smith M" |
| "Henry Myers" | "Myers H" | "Myers H" |
| "Barry Underwood" | "Underwood B" | "Underwood B" |
| "Sandy Jones" | "Jones S" | "Jones S" |

| | |
|---|---|
| LP | `GetToken_PROP_CASE_2 | ConstStr(" ") | GetToken_CHAR_1(GetToken_PROP_CASE_1)` |
| LP Latent | `TOK_30 | TOK_13 | TOK_39 | TOK_30` |

| Inputs | Outputs | LP Outputs |
|---|---|---|
| "January 15" | "jan 15" | "jan 15" |
| "febuary 28" | "feb 28" | "feb 28" |
| "march 1" | "mar 1" | "mar 1" |
| "October 31" | "oct 31" | "oct 31" |

| | |
|---|---|
| LP | `ToCase_LOWER(SubStr(1, 3)) | ConstStr(" ") | GetToken_NUMBER_1` |
| LP Latent | `TOK_11 | TOK_26 | TOK_17` |

| Inputs | Outputs | LP Outputs |
|---|---|---|
| "(321) 704 3331" | "321.704.3331" | "321.704.3331" |
| "(499) 123 3574" | "499.123.3574" | "499.123.3574" |
| "(555) 580 8390" | "555.580.8390" | "555.580.8390" |
| "(288)225 6116" | "288.225.6116" | "288.225.6116" |

| | |
|---|---|
| LP | `GetToken_NUMBER_1 | ConstStr(.)  | Replace_" "_.(SubStr(-8, -1))` |
| LP Latent | `TOK_17 | TOK_27 | TOK_24 | TOK_16` |

| Inputs | Outputs | LP Outputs |
|---|---|---|
| "Milk 4, Yoghurt 12, Juice 2, Egg 5" | "M.E." | "M.E." |
| "US:38 China:35 Russia:27 India:1" | "U.I." | "U.I." |
| "10 Apple 2 Oranges 13 Bananas 40 Pears" | "A.P." | "A.P." |
| "parul 7 rico 12 wolfram 15 rick 19" | "P.R." | ".." |

| | |
|---|---|
| LP | `GetToken_CHAR_1(GetToken_PROP_CASE_1) | Const(.)  |`
`GetToken_CHAR_-1(GetAll_ALL_CAPS) | Const(.)` |
| LP Latent | `TOK_39 | TOK_30 | TOK_6 | TOK_38 | TOK_30` |

Figure 8: Latent codes and programs found by Latent Programmer in string transformation tasks. Red denotes I/O where the predicted program mapped input to an incorrect output.

| Docstring | Program |
|-----------|---------|
| get an environment variable | ```python
def set_key(key, val, key_prefix=None):
    return environ.get(key, key_prefix)
``` |
| return a list of the words in the string s | ```python
def split(s, sep=None, maxsplit=-1):
    return s.split(sep, maxsplit)
``` |
| mean squared error function | ```python
def mean_squared_error(y_true, y_pred):
    return tf.reduce_mean(tf.square((y_true -
y_pred)))
``` |
| read a python file | ```python
def read_file(fname):
    f = open(fname)
    with open(fname, 'r') as f:
        f.seek(0)
        return f.read()
``` |
| pickle dump | ```python
def pickle_save(filename, data):
    with open(filename, 'r') as f:
        pickle.dump(data, f)
``` |
| takes a timedelta and returns the total number of seconds | ```python
def total_seconds(delta):
    return ((delta.microseconds + ((delta.days
* 24) * 3600) * (10**6))/(10**6))
``` |

Figure 9: Programs found by Latent Programmer in Python code generation dataset. Red denotes ares where the predicted program deviates from human code.

