# OpenReview forum: "Latent Programmer: Discrete Latent Codes for Program Synthesis"
_ICLR.cc/2021/Conference — Reject_

### Official Review · AnonReviewer2 · 2020-10-15
**Interesting paper, but very unclear and evaluation is lacking**

**Rating:** 3
**Confidence:** 4

**Review:**

### Summary ###
The paper addresses the problem of program synthesis from examples, and also evaluates on program synthesis from natural language descriptions.
The paper proposes Latent Programmer, an approach that employs an adapted VQ-VAE to predict a sequence of latent codes, and then generates the output program conditioned on those codes.
The evaluation shows improved results over straightforward LSTM and Transformer baselines.

Overall, I think that the approach is interesting, but the paper is very difficult to read and follow, and I am not sure that the evaluation is sufficiently extensive. I am thus voting for rejection at this time.

### Strengths ###
+ The proposed approach is a first application of VQ-VAE to program synthesis.
+ The models outperforms LSTMs and Transformers

### Weaknesses ###
- The paper is very unclear and difficult to follow
- It is unclear **why** the approach works
- It is unclear whether this is a direct application of VQ-VAE to program synthesis, or is there any novel insight.
- The evaluation is lacking comparison to additional baselines and datasets.

**Clarity** - the paper is very difficult to read and follow. For example, the first few pages contain obscure phrases like "discrete latent codes", "our sketches are comprised of a general latent vocabulary", "the semantics of the latent codes", "stop gradient operator", "sequence of tokens in latent space", "general vocabulary of discrete latent variables". While I could understand each individual word, it was unclear what exactly do the authors refer to.

The related work section is thorough and compares previous approaches to the proposed approach, but at this point in the paper, the reader has no idea what these comparisons refer to.
Only by the end of page 4 I had a general idea in mind of what the proposed approach does, and still, I couldn't perfectly match the description to Figure 2 (Was $Z'$ defined anywhere? What *exactly* do the authors mean by quantization and how coarse is it? how are the program and the I/O examples encoded?).

Additionally, there are plenty of smaller things that make the paper even harder to follow. For example: the caption of Figure 2 is uninformative; Equation (1) and its preceding text could simply say "nearest neighbor"; in Equation 2 it isn't even clear what is $x$, and what kind of encoder is $ec$, both architecturally and in terms of representation? Are the two Transformers "hierarchical" or just "two-level" or "two modules" (Section 4.1, all these phrases are used to describe these two transformers); the equations in Equation 3 are written from right-to-left. Section 4.1 says that "the plan is a sequence of tokens in latent space, denoted <TOK_1>, ..., <TOK_K>". If these are "tokens in **latent** space", can we just call them "a sequence of vectors", instead of using the word "token" in an unusual way?

**Why it works** - the paper does not give much intuition/explanation to **why** does the proposed approach work. The argument from the Abstract / Introduction that "discrete latent codes can learn a useful representation for search" is not very convincing, because eventually, the search is performed in the program space. So, the model does need to generate a sequence of tokens eventually. I think that maybe what the two steps of (1) generating the codes and then (2) generating the program given the codes do is provide more *diversity* of solutions, rather than better search.

**VQ-VAE** - before reading the paper, I was not familiar with VQ-VAE. The background in Section 3 was not sufficient and not clear enough. I understand that explaining a new background concept is challenging, but I still think that Section 3 could be written more clearly *without* taking more space.
For example, by saying "nearest neighbor" instead of multiple lines and equations, and without citing "van den Oord et al., 2017" three times on the same half a page.

Another issue is that I am not sure whether this paper simply applies VQ-VAE on program synthesis, or is there a novel adaptation following insights about programs?
Further, is the focus the VQ-VAE really needed, or can a standard VAE work as well (or even an AE)?

**Evaluation** - the evaluation presents impressive results, mainly compared to LSTMs and Transformers, which is an important comparison that is not always performed in all papers. However, it seems that there are not enough baselines to put the results in context. Additionally, the datasets could have been more standard: why generating a new dataset for string transformation? why taking such a small dataset for NL->python?

* Baselines - aren't there any recent baselines for string transformation, other than RobustFill? There is no other neural or non-neural available model for program synthesis from examples?
In NL->code, I am sure that there are other baselines other than Wei et al., 2019. For example:

1. Iyer et al., Mapping Language to Code in Programmatic Context, 2018
2. Iyer et al., Learning Programmatic Idioms for Scalable Semantic Parsing, 2019
3. Yin et al., "TRANX ..", 2018
4. Shin et al., "Program Synthesis and Semantic Parsing with Learned Code Idioms", 2019

and other semantic parsing papers.

* Datasets - in the program synthesis task, the authors created a new dataset. Can the authors use existing datasets, where previous work already tuned their baselines on?
In the code generation task, aren't there any other datasets, with hopefully more than 11K examples?

* Analysis - the authors do perform some analysis in Section 5.2, but it still feels that it is unclear why the model works. For example, can the VQ-VAE be simply VAE or AE? Are all the losses important? What I am looking for is a simpler architecture that achieves similar results using the same general idea.

### Questions to Authors ###
1. How can two Transformers have only 15M parameters? What would happen if RobustFill-Transformer used a full-sized Transformer-base?
2. Can we have an additional baseline that improves diversity in RobustFill, to reject the hypothesis that the VQ-VAE simply increases the diversity of solutions? I am not sure how.
3. The phrase "self-supervised" appears 5 times throughout the paper. Is it really self-supervised? The self-supervised part is that the encoding of the I/O examples is compared with the encoding of the program. Aren't the pair (program, specification), in fact, supervision?
4. Can the approach work with a code generator that does not generate the code as a sequence of tokens, such as Maddison & Tarlow (ICML 2014), Yin & Neubig (ACL 2017), Brockschmidt et al. (ICLR 2019), or Alon et al (ICML 2020)? or is the proposed approach limited to only "textual" code generators?

### Minor questions and comments ###
* Related work, paragraph2: "these works does not" -> "these works do not"
* It would help following if Figure 2 was in the same page that it is referenced
* It would be easier to read the appendix and refer it if the authors could append the appendix after the references, in the same PDF, instead of attaching it as a separate zip.

====== Post discussion comments ======

After reading all reviews, responses, and discussions, I still do not see evidence that the model has learned a "high-level plan", which is the main claim of the paper.

I agree that most deep learning models are not interpretable, but most papers do provide some qualitative/anecdotal/generality/strong empirical evidence to support their claims. In this paper, I do not see such strong evidence.

My main concerns:

1. Is it possible that the 2-stage search simply increases the diversity of solutions?
If there is an empirical improvement, I would like to understand the simplest explanation ("Occam's razor"). If the main contribution is diversity, I would expect the authors to spell it out clearly.

Further, if the main contribution is diversity, maybe there are much simpler and general ways to achieve diversity (e.g., diversity inducing versions of beam search), that can be applied to different architectures (i.e., not coupled with VQ-VAE).

I feel that my question "Can we have an additional baseline that improves diversity in RobustFill, to reject the hypothesis that the VQ-VAE simply increases the diversity of solutions?" was not answered by the authors.

2. The 2-stage search is a general approach, but the paper did not convince me that it is useful to settings beyond the FlashFill task, and for models other than the textual approach where programs are generated as text.

If the authors claim that their approach allows "high-level planning", I would expect to see that it works across different models / tasks / datasets / settings.

Minor: I do not agree with the authors that this is self-supervised. I think that the paper uses the term "self-supervised" incorrectly.

---

> ### Author Response · Authors · 2020-11-18
> **Response to AnonReviewer2 (1/2)**
>
> Thank you for your review. We would like to address the concerns you had below.
>
> (1) “...paper is very difficult to read and follow.”
>
> We revised the paper following your comments regarding the clarity of our work. Namely, we moved the related work section to after our method is introduced, and fixed your comments about components of our work that are hard to follow. We hope this makes our approach easier to understand.
>
> (2) “...does not give much intuition .. to why does the proposed approach work”
>
> We can try to give additional intuition for why we believe the two level search is helpful, and have updated the introduction of our revised work to make this more clear. To achieve a given task, often a programmer starts by specifying high-level pieces as a plan, and then filling in the details of each piece. For instance, a plan could be to “extract the first name, then the last initial", without working out full details about each step. We propose to use a sequence of tokens, called "discrete latent codes", to represent such plans. Instead of having a fixed dictionary of plans/codes, we let the model discover and learn what are the useful plans and how to convert a specification into latent codes. Being discrete, at inference time, we get advantage of employing combinatorial search techniques over a smaller space of possible high-level plans and then over programs conditioned on the code as a “plan”, in a two-level procedure.
>
> Although the model needs to generate a program eventually, we still argue that the top-level search can help to organize the lower-level search over programs. For example, the model could be confident that it needs to extract the last initial, but is less sure about whether it needs to extract a first name or not. By making a one-token change to the latent sequence, the search procedure can explore potentially many different alternative programs that do different things in the beginning. Whereas in traditional single-level search, the model would need to explore different multi-token prefixes of the alternatives. This is hard to achieve in limited budget beam search, which is notorious for producing a low diversity of solutions, as observed in NLP.
>
> Another significant advantage is in the ability of our two-level search to leverage structure. For example, in long programs where similar blocks are repeated often, our two-level search can first capture this structure, which would greatly reduce how much the low-level search over programs would have to do. This is most evident in the fact that our method synthesizes long, complex programs much better than baselines that only search over programs.
>
>
> (3) “Can the authors use existing dataset…”
>
> We use existing datasets and training sets.
> In the string editing domain, we follow the methodology of prior work of randomly generating programs from a fixed DSL (Devlin et al., 2017, Nye et al., 2019, Balog et al., 2020), so we are not aware of a publicly available, centralized dataset of string editing tasks. In the code generation task, we used the public dataset also used in Wei et al., 2019; we were also concerned with the size and quality, but ultimately chose this dataset because of its availability.
>
> (4) “What I am looking for is a simpler architecture that achieves similar results using the same general idea...”
>
> This is an excellent point. In the revised paper, we added two additional ablative baselines that replaced the VQ-VAE component with a generic AE and a VAE. The results are in the revised paper, and summarized in a table below:
>
> Method: B=1, 10, 100 accuracy
>
> RobustFill [Transformer]: 47, 51, 61
>
> Latent RobustFill [AE] : 47, 50 , 60
>
> Latent RobustFill [VAE]: 46 , 51, 62
>
> Latent Programmer: 51, 57, 68
>
> We are also working on comparing our method to SketchAdapt [1], which first generates a program sketch with holes then fills the holes in via enumerative synthesis. We will add another comment if we are able to get results during the rebuttal phase.
>
> [1] Nye et al. (2019) https://arxiv.org/pdf/1902.06349.pdf

---

> > ### Author Response · Authors · 2020-11-18
> > **Response to AnonReviewer2 (2/2)**
> >
> > (5) “What would happen if RobustFill-Transformer used a full-sized Transformer-base?”
> >
> > We wanted to keep the number of trainable parameters consistent with the original RobustFill model, so we chose our embedding and hidden dimensions accordingly. We performed additional experiments in Section 5.2 where the model size was increased, but unexpectedly saw diminishing returns, so we do not anticipate using a full-sized Transformer would greatly impact performance.
> >
> > (6) “Can we have an additional baseline that improves diversity in Robustfill…”
> >
> > This is non-obvious to do, but we imagine a solution can involve adding a diversity penalty in the beam search. However, we view this as an orthogonal research problem.
> >
> > [1] Vijayakumar et al. (2018) https://arxiv.org/pdf/1610.02424.pdf
> >
> > [2] Kool et al. (2019) https://arxiv.org/pdf/1903.06059.pdf
> >
> > (7) “Is it really self-supervised?”
> >
> > We use the term “self-supervised” because we effectively have two supervised objectives, one that depends on the output of the other. We do not feel strongly about whether to call it self-supervised or supervised and would be willing to change it if strong opinions did exist.
> >
> > (8) “Can the approach work with a code generator that does not generate the code as a sequence of tokens.”
> >
> > Changing the code generation from an autoregressive sequence of tokens is an orthogonal complementary direction. Our goal was to compare two-level search in neural program synthesis against single-level search, and chose the generation framework of the baseline RobustFill to do so. A different decoder (i.e. Tree-LSTM) can be co-opted into the model architecture instead of a Transformer decoder.

---

> > > ### Comment · AnonReviewer2 · 2020-11-19
> > > **AnonReviewer2**
> > >
> > > Thank you for your response.
> > >
> > > I still think that the evaluation is not convincing. Following the RobustFill+VAE results, now I am even more doubtful about the claims that the two-step search is the main contributing factor.
> > >
> > > >To achieve a given task, often a programmer starts by specifying high-level pieces as a plan, and then filling in the details of each piece...
> > >
> > > >we still argue that the top-level search can help to organize the lower-level search over programs...
> > >
> > > >Another significant advantage is in the ability of our two-level search to leverage structure...
> > >
> > > These are interesting hypotheses, but I don't see **any evidence** that this is what the model had actually learned.
> > > Further, as the RobustFill+VAE results show below, this two-level search is *not* the main contribution source.
> > >
> > > > We use existing datasets and training sets... we follow the methodology of prior work of randomly generating programs from a fixed DSL
> > >
> > > I see these two sentences as contradicting. If the authors generated programs, this is not an existing dataset. Can the authors take the exact same generated programs as (Devlin et al., 2017, Nye et al., 2019, Balog et al., 2020)?
> > >
> > >
> > > > In the code generation task, we used the public dataset also used in Wei et al., 2019; ; we were also concerned with the size and quality, but ultimately chose this dataset because of its availability.
> > >
> > > It would be more convincing if the authors used a larger dataset, as detailed above.
> > >
> > > > In the revised paper, we added two additional ablative baselines that replaced the VQ-VAE component with a generic AE and a VAE.
> > >
> > > What are the conclusions from this experiment?
> > > It seems to me that the RobustFill baseline performs equally to RobustFill+VAE.
> > > So, the quantization (VQ) is what gives the highest gain, and not the two-level search?
> > >
> > > > We wanted to keep the number of trainable parameters consistent with the original RobustFill model
> > >
> > > I am not sure that keeping the number of trainable parameters consistent with a model from 2017 is a realistic assumption.
> > >
> > > > We performed additional experiments in Section 5.2 where the model size was increased, but unexpectedly saw diminishing returns
> > >
> > > To me, it doesn't look like diminishing returns actually. The accuracy seems to be increasing linearly in Figure 3(a).
> > >
> > > > This is non-obvious to do, but we imagine a solution can involve adding a diversity penalty in the beam search. However, we view this as an orthogonal research problem.
> > >
> > > I definitely understand that this is a non-obvious thing to do, but unfortunately, the current evaluation makes it very difficult to pinpoint the most contributing factors. There are several components and the relation between them is unclear.
> > >
> > > > We use the term “self-supervised” because we effectively have two supervised objectives, one that depends on the output of the other... would be willing to change it if strong opinions did exist.
> > >
> > > I do not have strong feelings about it either, I simply think it is an incorrect claim.
> > >
> > > >Changing the code generation from an autoregressive sequence of tokens is an orthogonal complementary direction
> > >
> > > I agree that it might not be feasible within this discussion period. But it would strengthen the generality claim of the two-level approach. If the authors claim that the general two-level approach is what matters, it will be much more convincing to show that it works across different synthesis/generation approaches, and not only within a specific implementation.

---

> > > > ### Author Response · Authors · 2020-11-20
> > > > **Response to AnonReviewer2's Concerns**
> > > >
> > > > Thank you for the thoughtful reply. You raised some important concerns that we would like to address below.
> > > >
> > > > (1) “The RobustFill+VAE results show… two-level search is not the main contribution source.”
> > > >
> > > > Respectfully, we believe that this does not follow. The main claim in our paper is that two-level search can specifically help when we use a learned discrete representation, because we can use combinatorial search techniques like beam search at the top level. The RobustFill+VAE ablation directly shows that a different, natural method for doing two-level search in continuous space (sampling at the top level) does not work as well as our proposed two-level search in discrete space. Thus, we would argue that these results strengthen the arguments in our paper.
> > > >
> > > > (2) "So, the quantization (VQ) is what gives the highest gain, and not the two-level search?"
> > > >
> > > > This is a reasonable concern, but we are not sure how it is possible at a conceptual level to do VQ without two level search. If you do VQ, then a latent sequence must be generated at test time somehow. Then it seems to us that any method for doing so could be interpreted as two level search. (RobustFill+VAE is the other ablation, two level search without VQ, and that works worse).
> > > >
> > > > (3) What is the evidence that the two-level search helps?
> > > >
> > > > Great question. We suggest that this evidence appears in Figure 5, where we vary the amount of search done at the top level vs the second level. In the L=1 row, we generate one latent sequence greedily, then do beam search of size 10 over programs. For L=3, we generate 3 latent sequences by beam search, and then for each of these, we generate 3 programs, yielding 9 programs in total. We see that L=3 outperforms L=1, which we take as direct evidence that the two-level search helps.
> > > >
> > > > (4) “Can the authors take the exact same generated programs…”
> > > >
> > > > The methodology from prior work is to randomly sample new programs from the DSL at each minibatch during training, and to sample new programs during evaluation. This is what Devlin et al (2017) do, as well as the other work that we cite (Nye et al. 2019, Balog et al. 2020). As these programs are sampled online, these authors do not make the sampled programs publicly available.
> > > >
> > > > (5) "To me, it doesn't look like diminishing returns actually."
> > > >
> > > > That's a good point, we agree. That said, we would say that: 1. given the slopes of the learning curves in Figure 3(a), it is not at all clear to us that training larger models would provide more convincing evidence about the difference between the methods, and 2. our largest models are ~20M parameters, which is a similar order of magnitude as recent work in the program synthesis (Balog et al. 2016, Devlin et al. 2017, Nye et al. 2019, Balog et al. 2020). Model sizes are different in program synthesis vs NLP.

---

### Official Review · AnonReviewer1 · 2020-10-27
**nature of VQ-VAE?**

**Rating:** 4
**Confidence:** 4

**Review:**

This paper proposes a VQ-VAE approach for program synthesis (generating a program from specifications, either input-output pairs or natural language description). Generally speaking, a VQ-VAE learns an autoregressive discrete latent code in addition to traditional Seq2Seq learning, and perform beam search on both latent code and output programs.

Experimental results show that the model outperforms three baseline systems on two tasks.

---
I have major concerns on the nature of the VQ-VAE model.


1) First of all, this paper heavily relies on Kaiser et al. (ICML'2018), except that the decoder of this paper is autoregressive and that this paper proposes beam search on the latent sequential discrete codes.

However, this paper has very light citation on Kaiser et al. (2018). The authors should be more honest about previous work and make direct comparison on the difference. What is taken from previous work? What is an adaptation? What is an extension?

The current writing shows that this paper has a heavy development on the model, when in fact, it's mostly taken from previous work.

2) The author claims that VQ-VAE serves as a discrete bottleneck. However, I strongly disagree with this.

The decoder in this paper is well aware of the input by "TransformerDecoder(Y', E)" in Eq 4, where the E = TransformerEncoder(X).

So, the decoder can just learn from the input X and disregard the VQ-VAE space, despite a few semantic losses imposed on the latent code (latent prediction and end-to-end in Eq 5). Since the VQ-VAE latent space is in addition to a traditional Seq2Seq training, it cannot serve as a bottleneck/regularization.

This is known as the "bypassing phenomenon" in previous work:

Bahuleyan et al., Variational attention for sequence-to-sequence models, 2018.

The authors may want to explain why their VQ-VAE would not suffer from the bypassing phenomenon.

Note: the bypassing phenomenon is actually different in  Kaiser et al. (2018). Their decoder is non-autoregressive, so their sequential discrete latent space can provide autoregressive information. But in this work, the decoder is autoregressive, which can simply learn from X directly.

3) What's the real benefit of modeling the discrete latent codes by VQ-VAE? A natural way of having real discrete bottleneck is by reinforcement learning. There lacks comparison and discussion.

Note: we all know RL systems are difficult to learn, but the auxiliary losses is Eq 5 can all applied to RL, too. You may also do pre-training or relaxations for RL.

4) The latent codes are generated in an autoregressive fashion. During training, the number of latent codes is ceiling(T/(2^l)). But how do you know the number of codes during test? Did you include an EOS token for such autoregressive generation? If yes, how easy is it to learn the precise semantics of EOS without direct supervision signal?

5) There is no quantitative analysis on the learned code. While there is an example, it is inadequate. We have no measure on how typical is the shown example.


---

I also have major concerns on experiments.

6) The evaluation metrics are peculiar. For example, distance n-grams are used to evaluate diversity. We understand diversity is important for natural language generation, but why do we need diversity for program generation?

The BLEU is computed by the best BLEU score among the output beams, but increasing the beam size may not improve top-beam performance.

7) Baseline models are inadequate. For the example-to-program generation, the authors only compared with Seq2Seq with LSTM or Transformer. There has been other efforts on search-based program synthesis, for example, Balog et al. (2017, 2020). I'd like to see a comparison, and what's the further improvement when they are combined (as claimed by this paper)?

In code generation from description, the authors only compared with two models in Wei et al., 2019 and two variants of Seq2Seq. But there could be more benchmarked datasets, like Hearthstone, spider, and other semantic parsing datasets in the old days.

---

Minor:

The exponential moving average is not proposed in Roy et al. (2018). It's proposed in Appendix A of the original VQ-VAE paper.

---

In short, the discrete latent space beam search appears to be some interesting idea. But I have concerns on the soundness of this paper.

It is also noted that there's no code or output available.

---

> ### Author Response · Authors · 2020-11-18
> **Response to AnonReviewer1 (1/2)**
>
> Thank you for the detailed feedback. We would like to address some important concerns that you raised with the work.
>
> (1) “this paper heavily relies on Kaiser et al.”
>
> In our view, the novelty of our work lies not in the model architecture (we agree those differences are minor), but in how we use the latent codes to perform search. We propose a two-level search, one over a high-level, compact, and learned latent space and the other over the larger program space, in order to improve accuracy in program synthesis tasks. This involves learning a discrete latent code so that beam search can be used on both levels of search. To our knowledge, this is a novel strategy for learning to search.
>
> Thus, the distinguishing feature of our work is in casting program synthesis as a two-level search problem; using a VQ-VAE (as Kaiser et al.) enables us to learn the first level. We agree that the mechanism that we use to learn latent codes relies heavily on Kaiser et al. However, we note that their work work leverages a VQ-VAE to learn a discrete latent state in order to reduce the decoding latency for text generation tasks, not as a way of better exploring the output space during search.
>
> We tried to adequately cite the Kaiser et al. work (e.g. we cited it in the introduction) but perhaps we could have been more explicit in detailing the differences between their work and ours. We added a more detailed discussion in the related work section of our revised version, and clarified that the key contribution of our work is to propose two-level search for program synthesis.
>
> (2) “the decoder can just learn from the input X and disregard the VQ-VAE space...”
>
> Thanks for the reference. We were not aware of the term "bypassing phenomenon" but we also observed this phenomenon in early versions of our method, where the latent codes were ignored during decoding. To avoid this, we added a pre-training step where the true program is initially passed instead of the latent states so reasonable gradients can be passed through the latent states in the decoder. After making this change, we know that the decoder is not disregarding the latent space because it significantly outperforms Robustfill empirically (57% with latent vs 51% without). We explain this at the end of Section 3.2 and have also added the reference in the revised version.
>
> (3) “A natural way of having a real discrete bottleneck is by reinforcement learning…”
>
> RL is definitely an appropriate framework, as program synthesis can (like many other applications) be cast as an MDP. We chose a supervised approach simply because there is little prior work in applying RL to program synthesis. We feel that successfully applying RL would be an independent research direction, as it would involve handling additional concerns such as computational overhead, reward sparsity, etc. We view RL as complementary to this work, and can be used in lieu of VQ-VAEs to train and search the latent space, and in finetuning the model.
>
> (4) “how do you know the number of codes during test?”
>
> We included an EOS token that is used to end latent sequences during inference. Following standard beam search conventions, we included a brevity penalty to favor shorter sequences. Empirically, we noticed that the latent codes during inference are also observed to be proportional to the length of the synthesized programs, even though the length of the program is not known beforehand. It makes sense to us that the latent predictor would produce output sequences of similar length during inference as it saw during training.
>
> (5) “There is no quantitative analysis on the learned code.”
>
> This is an important point. Please see (2) of our response to AnonReviewer 3 where we discussed interpretability of the latent code.
>
>
> (6) “Why do we need diversity for program generation?”
>
> We are concerned with a system that outputs a list of programs, where all programs in the list can be evaluated (not necessarily just the top-1). In such a scenario, it is often advantageous for the programs in the list to be relatively diverse so that some program in the list is more statistically likely to be correct. This is why we additionally measure diversity, and also why our evaluation chooses the program with the maximum BLEU score.

---

> > ### Author Response · Authors · 2020-11-18
> > **Response to AnonReviewer1 (2/2)**
> >
> > (7) “Baseline models are inadequate.”
> >
> > The crux of our work is in showing that two-level search over latents and programs can outperform traditional search over just programs. We kept the search algorithm (beam search) constant across all baselines; many works, such as Balog et al., propose alternative methods of search than beam search that can easily be co-opted.
> > However, we did add two additional ablative baselines that replaced the VQ-VAE component with a generic AE and a VAE to the revised version of the paper. We are also working on comparing our method to SketchAdapt [1], which first generates a program sketch with holes then fills the holes in via enumerative synthesis. We will post a new comment if we are able to get results during the rebuttal phase.
> >
> > [1] Nye et al. (2019) https://arxiv.org/pdf/1902.06349.pdf

---

> > ### Comment · AnonReviewer1 · 2020-11-20
> > **thanks for the response**
> >
> > 1, 2, 4, 6 are mostly resolved. 7 is still ongoing, but if your primary goal is two-stage BS regardless of  code synthesis, I'd like to see its application in other tasks, like diverse text generation.
> >
> > I don't think the response to (3) is valid. There's no difference between VQ-VAE and the RL in my mind in terms of supervision level. What I suggest is to perform RL of the discrete latent codes by using output as the reward. The generation of the code is still supervised, the learning of latent code is by RL.
> >
> > The response to (5) is also weak. Sec 5.2 is a hyperparameter tuning, but it doesn't explain what the codes are. Probably you can do a clustering or some correlation analysis between latent codes and output tokens?

---

### Official Review · AnonReviewer3 · 2020-10-28
**Interesting idea and convincing results**

**Rating:** 7
**Confidence:** 3

**Review:**

Summary: This paper proposes a two-level hierarchical program synthesizer, Latent Programmer, which first predicts a sequence of latent codes from given input-output examples, and then decodes the latent codes into a program.  The sequence of latent codes can be viewed as a high-level synthesis plan, guiding the subsequent low-level synthesis. Latent Programer significantly outperforms RobustFill on string manipulation tasks and achieves state-of-the-art results on Python code generation tasks.

Quality: The paper presents a novel program synthesis idea and the evaluation is promising and convincing.

Clarity: The writing provides enough background and explains the main idea in a very clear manner.

Originality: The application of Vector Quantized Variational Autoencoder for a two-level hierarchical synthesis is quite novel.

Significance: This work shows that a promising hierarchical learning approach for program synthesis. Its effectiveness motivates many future explorations in this direction.


Questions:

Q1: Why Python Code generation tasks use BLEU as the metric, rather than functional correctness?

Q2: Latent codes are motivated as a "high-level plan"? Do you observe certain interpretability of latent codes?

Q3: Since the lengths of synthesized programs are different for different tasks, it might be good to have a task-specific length of latent codes. The authors do show that varying the length of latent codes could affect performance. Is the length of latent codes (always) proportional to the length of synthesized programs?

---

> ### Author Response · Authors · 2020-11-18
> **Response to AnonReviewer3**
>
> Thank you for your review. We address the questions that you had below.
>
> (1) “Why … use BLEU as metric, rather than functional correctness?”
>
> This choice follows previous work (Wei et al 2019). We agree that in principle functional correctness is a better metric, and we use functional correctness metric for evaluating synthesized programs for the string editing task. The reason why we use BLEU for the Python code generation task is because the dataset consists of publicly scraped functions on Github. These functions are often not executable due to several reasons such as missing dependencies in the library they were obtained from, having complex input objects as arguments, or absence of test cases. Therefore, we report the BLEU score instead, following Wei et al.
>
> (2) “Do you observe certain interpretability of latent codes?”
>
> One of the strengths of our work is that individual tokens in the discrete latent code can have arbitrary meaning, allowing the latent representation to be very rich and expressive. However, this is also a weakness, as we did not perform any grounding on the latent code to induce interpretability i.e. make individual tokens explicitly map to high-level API calls. The most we could do is provide a qualitative analysis via examples in the main paper and appendix. From the examples, we noticed that in programs with a repetitive structure, the predicted latent sequence would often have repeated tokens; however, in less-obviously structured programs, the latent sequence became difficult to interpret. This is an important point though and we have added a discussion to describe this more thoroughly in the revision in Section 5.2.
>
> (3) “Is the length of latent codes … proportional to the length of synthesized programs?”
>
> During training, we always make the ground truth latent code proportional to the length of the programs. This was done mostly for simplicity and could be improved in future work. During inference, however, the estimated latent codes have no restriction on length, as the latent predictor will continue to generate tokens in the latent sequence until an EOS token is generated.

---

### Official Review · AnonReviewer4 · 2020-10-28
**Promising neural program synthesis approach.**

**Rating:** 7
**Confidence:** 4

**Review:**

Edit: I have increased my score to 7.


This paper introduces a novel program synthesis system called the Latent Programmer, which uses discrete latent codes as a representational scheme to solve program synthesis problems in two domains: string transformations from examples and code generation from language descriptions.

Strengths:

-The paper is relatively clear.

-The approach is novel.

-The results seem to support the claim that this model outperforms baselines (although it would help to report the results of multiple runs with standard error).

-The relative simplicity of the approach is a plus; it doesn't seem that it would be terribly difficult for a researcher to adopt this technique to a new problem.

Weaknesses:

I think the baselines/ablations could be more complete. For example, it seems that the gains over the RobustFill baselines could be due to any of 3 factors: 1) use of discrete representations 2) the use of an autoencoding loss, or 3) the ability to search through latent representations at test time.

Unless I'm mistaken, compared to LP, the transformer RobustFill baseline differs in terms of both (1) and (2): RobustFill does not use discrete latent codes, and it does not use the autoencoding or latent prediction losses. As written, the paper seems to assume that (1) is the primary reason for the performance difference ("[the transformer RobustFill model] can also be considered of an ablation of our LP model without latent codes]"). However, I think these two factors need to be better disentangled, in order to determine which contributes most to the performance. Can a RobustFill model be trained with an additional auto-encoding loss, so that its loss function is more analogous to LP? Similarly, how might a continuous latent variable model, such as a VAE, perform on the string editing tasks?

Similarly, it seems there is evidence that (3) is an important factor: in Figure 5, when doing a beam search of size 10, but only searching in the decoder space and keeping the latents fixed (L=1), the performance seems identical to the transformer RobustFill baseline. LP seems to beat baselines with B=1. What are the results for B=100 and L=1?

I think that disentangling these factors would really strengthen the paper, and could also be of large value to the neural program synthesis community.

Summary:

I think this is an interesting line of work with promising results. However, I do think that a baseline which uses an autoencoding loss but does not use discrete latent codes is an important ablation to perform. I therefore recommend a weak accept, and I'd be willing to raise my score if my concerns about baselines were addressed.

---

> ### Author Response · Authors · 2020-11-18
> **Response to AnonReviewer4**
>
> Thank you for your review. We would like to address the points you raised below.
>
> (1) "Three factors..."
>
> The reason that we chose to use discrete latent states (factor 1 in your review) is precisely because this makes it easier to search through latent factors at test time (3) using combinatorial search. So we believe that (1) and (3) are strongly coupled with each other. Searching over a continuous latent representation is much more challenging to do; though it has recently been done for control in robotics, it is unclear how the methods generalize to program synthesis [1].
>
> (2) "only searching in the decoder... seems identical to the transformer RobustFill baseline"
>
> As for why the L=1 case of our method performs similarly to the RobustFill baseline, our intuition is that if no search is performed over the latent representation, our proposed two-level search reduces to a single level, which is essentially similar to what the RobustFill baseline does. We imagine that the B=100 and L=1 base would also perform as well as RobustFill.
>
> (3) "disentangling these factors would really strengthen the paper..."
>
> Thanks for the suggestion of disentangling these factors to clarify the advantages of our proposed approach. We performed additional experiments to disentangle the use of discrete latent states from the use of an autoencoder (or (1) and (2) in your review). As you suggested, we have added two additional ablative baselines that replace the VQ-VAE of the Latent Programmer with either a generic AE or a VAE. The results are in the revised paper, and are also summarized in below:
>
> Method: B=1, 10, 100 accuracy
>
> RobustFill [Transformer]: 47, 51, 61
>
> Latent RobustFill [AE] : 47, 50 , 60
>
> Latent RobustFill [VAE]: 46 , 51, 62
>
> Latent Programmer: 51,  57, 68
>
> Both those baselines have a similar autoencoding loss but perform similarly to the Robustfill baseline; this is because we cannot perform search over the latent representation.
>
> [1] Watter et al. (2015) http://papers.neurips.cc/paper/5964-embed-to-control-a-locally-linear-latent-dynamics-model-for-control-from-raw-images.pdf

---

> > ### Comment · AnonReviewer4 · 2020-11-24
> > **Baselines**
> >
> > Thank you very much for running the baselines. It does seem more clear now that the autoencoding loss is not the primary factor contributing to the results. It's still interesting to note that the B=1 results of the LP model exceed the performance of baselines, so it seems that two-level search is not the only factor contributing to better performance.
> >
> > Despite remaining questions, I believe the approach is interesting and could be valuable to the program synthesis community. I will therefore increase my score to 7.

---

### Author Response · Authors · 2020-11-18
**Updates to the Paper**

We would like to thank the reviewers for their insightful feedback. Following the advice of several reviewers, we provided an illustrative motivation of our two-level search idea in the introduction, clarified some aspects of our method, improved the comparison to prior work, and added two additional ablative baselines to show the importance of having a discrete latent space and being able to search over it via beam search. For our two baselines we replaced the VQ-VAE component of the Latent Programmer with a generic AE and a VAE. In the former only one latent sequence can be decoded per task, and in the later latent sequences can be sampled but not searched over.  The results are in the revised paper, and are also summarized below:

Method: B=1, 10, 100 accuracy

RobustFill [Transformer]: 47, 51, 61

Latent RobustFill [AE] : 47, 50 , 60

Latent RobustFill [VAE]: 46 , 51, 62

Latent Programmer: 51, 57, 68

We are also working on evaluating our method in the same domain as SketchAdapt [1], which first generates a program sketch with holes then fills the holes in via enumerative synthesis. Among prior work in neural program synthesis, we think that the method proposed in that work is most similar to our two-level search. We will add another comment if we are able to get results during the rebuttal phase.

[1] Nye et al. (2019) https://arxiv.org/pdf/1902.06349.pdf

---

### Author Response · Authors · 2020-11-24
**Added Baseline to the Paper**

We have updated our submission with a comparison to SketchAdapt [1], which is a recent work that also proposed a form of two-level search for program synthesis. Instead of using a learned latent vocabulary, SketchAdapt expands the program vocabulary with an additional HOLE token; the top-level search generates a program with holes, and the bottom-level search does enumerative synthesis of partial programs to fill in the HOLE token. The paper uses a slightly modified DSL that is more amenable to enumerative synthesis. We had trouble getting SketchAdapt to perform as well in our DSL implementation, so instead, we ran our Latent Programmer method on data generated by SketchAdapt's modified DSL (using code by the authors), and compared our results to the ones in Figure 3 of the paper. We were able to improve upon SketchAdapt even in this modified DSL:

Method: B=50, 100 accuracy

SketchAdapt: 63, 64

Latent Programmer: 64, 67

[1] Nye et al. (2019) https://arxiv.org/pdf/1902.06349.pdf

---

### Decision · Program_Chairs · 2021-01-07
**Final Decision**

**Decision:**

Reject

**Comment:**

The paper tackles program synthesis using a discrete latent code
approach, enabling two-level beam search decoding. The approach is well
motivated, as program synthesis requires high-level choices that affect long
subsequences of the output, and discrete codes are amenable to heuristic search.
Empirical results show improvements over methods with no latent
variables and methods with continuous latent variables. However, the review process reveals that some of the claims made about the nature and necessity of the discrete latent codes is not sufficiently justified by the current analysis. With borderline assessments in a very competitive venue, *I cannot recommend acceptance*. I would like to encourage the authors to pursue this direction further, shedding more light on the nature of the latent representations learned as an interpretable planning mechanism.

The discussion was rich and surfaced a lot of concerns and issues with
the paper, that I strongly encourage the authors to take into account.  After
the author responses and internal discussion, many initial concerns
were settled, but some remain. The two main concerns raised are: (1) that the
paper makes overly ambitious claims about high-level planning which is not
backed by an analysis of the latent codes themselves, and (2) that the
improvement may be due to increased generation diversity (which could be
possible in continuous LV models too) rather than meaningful
high-level planning. I believe that after clarifying such remaining loose ends,
this work would be of great interest to both the field of program synthesis as well as the discrete
representation learning community.